# Smart Products Enable Smart Regulations—Optimal Durability Requirements Facilitated by the IoT

**Moritz-C. Schlegel \***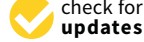**, Claudia Koch**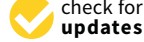**, Mona Mirtsch and Andrea Harrer**

Federal Institute for Materials Research and Testing (BAM), 12205 Berlin, Germany; claudia.koch@bam.de (C.K.); mona.mirtsch@bam.de (M.M.); andrea.harrer@bam.de (A.H.)
**\*** Correspondence: moritz-caspar.schlegel@bam.de

**Abstract:** The challenges and opportunities linked with IoT have been intensively discussed in recent years. The connectivity of things over their entire life cycle and the smart properties associated with it provide new functionalities and unprecedented availability of (usage) data. This offers huge opportunities for manufacturers, service providers, users, and also policymakers. The latter may impact policy areas such as the regulations on resource and materials efficiency under the Ecodesign Directive 2009/125/EC. With the general approach as it is practiced today, legal requirements are usually set for entire product groups without considering the products individually, including user behavior and environmental conditions. The increasing number of smart products and the growing availability of product data are sparking a discussion on whether these requirements could be more product and application-specific. This paper presents a method for calculating the economically and ecologically optimal durability of a product. It allows determining the point in time when a product should be replaced by combining consumer data with product design data. This novel approach could contribute to making product regulation more flexible and possibly more efficient. In this context, fundamental challenges associated with smart products in policymaking are also discussed.

**Keywords:** policy making; Ecodesign; IoT; material efficiency; resource efficiency; connectivity; big data; smart home

---

## 1. Introduction

Digitization is fundamentally changing both everyday life and the economy. The development of the Internet of Things (IoT) is accelerating, and the number of connected things was expected to rise from 8.4 billion in 2017 to more than 20 billion by 2020 [1]. The term IoT, first established by Kevin Ashton in 1999 [2], describes "an ecosystem in which applications and services are driven by data collected from devices that sense and interface with the physical world" [3,4]. Nowadays, also referred to as 'The Internet of Everything' [5], it describes a phenomenon where processes, people, data, and everyday objects are connected to the Internet to achieve specific goals.

Smart, connected things consist not only of conventional physical components but also of "smart" ones (e.g., sensors, controls, software) and connectivity components (e.g., ports and protocols). These smart, connected products have advanced capabilities and functions and generate an immense amount of new (and sensitive) data [6]. The availability, use, processing, and analysis of this big data offers unprecedented opportunities [3,7–12]. The level of feedback that producers can now receive—in real time—provides opportunities for product innovations and customized services. Further, it has implications for overall lifecycle management, and specifically for the monitoring of energy efficiency and replacement management of these smart products-with possible implications also for product regulation. By 2021, connected home appliances, such as connected white goods (refrigerators, washing machines, dishwashers, etc.) will account for almost half of all machine-to-machine connections, demonstrating the relevance of IoT in consumers' everyday lives [3]. The regulation of these white goods in Europe has been established by

---

the Ecodesign Directive as the legal basis for mandatory requirements for energy-related products entering into the European Market and/or being put into service [13]. Hence, the Directive distinguishes between product groups, rarely considering the different types of applications and technologies. The availability of huge amounts of product-related information throughout the products' life cycle under the IoT opens up new possibilities to achieve more flexible and individual product regulation.

In this paper, we discuss the potential of IoT in the context of regulation. Section 1.1 introduces the concept of IoT in smart homes and its technical and economic implications. In Section 1.2, the paper further presents the EU Ecodesign Directive and shows how product regulation could benefit from IoT and the resulting availability of huge amounts of data. In Section 2, a calculation method is introduced based on big data that considers ecological and economic aspects. Since this novel approach relies basically on the availability of the data generated and transmitted by the installed and used IoT products, Section 3 discusses some of the technological and legal challenges associated with this data. The discussion in Section 4 specifically addresses concerns and status in terms of data privacy and security.

### 1.1. IoT-Enabled Product Regulation

Nowadays, also referred to as "The Internet of Everything" [5], IoT is discussed and applied in industrial contexts, e.g., in connected cars, in healthcare, in smart cities, smart grids, and—as discussed here—in smart homes. The number of smart home appliances was estimated to grow from 1 million units sold in 2014 to over 223 million worldwide in 2020, including 11 million smart large cooking appliances, 17 million smart dishwashers, 120 million smart refrigerators, 131 smart washing machines and smart dryers, and 186 million smart air conditioners [14]. This tremendous growth is driven by many factors, such as more technology-savvy consumers (also due to the broad adoption of smartphones and the accordant familiarity with the technology) and the increased availability of connected devices [14]. But technological advancements, such as Artificial Intelligence or voice control, are also spurring development [15]. Since smart devices are considered more energy-efficient, the potential for saving energy also accelerates consumer acceptance [14,16]. At the same time, enormous amounts of data are generated using smart devices, e.g., on their status and functionality and information about their environment [17]. For instance, an IoT-enabled washing machine could provide information on the extent of use, failures, changes in performance, maintenance needs, etc. The related data enable the manufacturer to analyze and verify how often certain products need to be repaired and maintained and to explore correlations between usage profiles and failures [15]. Manufacturers and vendors typically rely on customers' information and feedback to learn about product failures [18]. However, the information about the use and end-of-life phases is inaccurate, incomplete and not real time and, therefore, preventing a closed loop where product designers and manufacturers could continuously use the data to improve the design, manufacture, use, and end-of-life handling of products [19,20]. This could potentially improve performance and ensure energy and resource-efficient operation [20].

Demestichas and Daskalakis [21] review literature on the roles of information and communication technologies (ICT) to support a circular economy (CE), identifying big data as one of the most popular CE enablers. In their systematic literature review, Nobre and Tavares [22] only found very few CE studies (<0.25%) that specifically consider the application of IoT and/or big data, confirming a yet "low amount of research on the field of environmental sustainability with the support of information systems" [22]. However, their analysis reveals an increasing scholarly interest over the last years [22].

In one of the few existing studies, Främling, Holmström, Loukkola, Nyman and Kaustell [20] used a connected smart refrigerator to demonstrate how continuously collected data on energy consumption, operating status and maintenance requirements are used to ensure energy and resource-efficient operation. In their study, it was possible to operate the refrigerator in an energy-efficient manner and to detect and alert abnormal conditions (e.g., open doors) in real time. For maintenance purposes, error logs could be

analyzed, and information on required spare parts could be provided. Status monitoring made it possible "to estimate its residual value to take timely decisions on when it has reached the end of its environmentally sustainable lifetime" [20]. However, such information on degradation and maintenance requirements is not only valuable for the producers and other economic stakeholders along the value chain.

Li et al. [23] discuss the role of IoT for Ecodesign in consumer electronics, pointing up potential enhancements, e.g., for saving resources. Askoxylakis [24] presents a framework for pairing CE and the IoT, emphasizing the potential to augment resource productivity, e.g., through monitoring and data management along the life cycles of products. How the IoT can offer new solutions to product lifecycle management is presented by Menon et al. [25], focusing on the role of specific platforms with "the ability to access, manage and control product-related information across various phases of lifecycle" for augmented insights into the condition of the products and potential need for maintenance [25].

Considering the growing regulatory pressure towards more sustainability, Zhang et al. [26] discuss how IoT technologies could enhance lifecycle information management in the automotive industry, providing information at all life stages and thus improving the recycling and recovery rate [26]. The availability of IoT data may also be relevant for regulation itself. Meanwhile, the role of IoT for smart government is broadly discussed [27]. However, there is not yet a systematic and strategic use of IoT and effective exploitation of its potential [28]. Metallidou et al. [29] investigated how the IoT can impact the efficiency of smart cities, highlighting improved energy performance certificates for smart buildings demanded by recent European legislation.

Our paper goes further, proposing an IoT-enabled calculation method for the economically and ecologically optimal durability of a product that can be used for a more flexible and effective Ecodesign regulation. We show how IoT data could contribute to higher flexibility and specificity for individual appliances and technologies in the framework of the EU Ecodesign Directive, as addressed in more detail in the next subsection.

## 1.2. Towards a More Flexible Regulation for Energy-Related Products

The Ecodesign Directive 2009/125/EC has established a framework setting Ecodesign requirements for energy-related products [13]. Since "Energy-related products account for a large proportion of the consumption of natural resources and energy" within the European Union, the Directive strives to encourage improvements in the overall environmental impact of these products. Although the Directive primarily addressed energy efficiency, the integration of resource efficiency has recently been discussed [30,31] and addressed in several political initiatives [32–35]. In addition to the focus on energy efficiency and the use phase, requirements are set only for product groups and not for individual technologies and applications. As outlined in the previous subsection, smart products' connectivity allows more information to be obtained and analyzed about the product's usage profile and environment. In addition, online monitoring of product performance parameters leads to better diagnostics. The need for maintenance, repair action, or the entire durability of every single product can be more accurately predicted. The obtained data can be used to calculate the point of time when it is more appropriate to replace a product than to maintain or repair it, covering both perspectives, economic (lower energy costs and costs for repair and maintenance) and ecological (lower consumption of energy and raw materials). This is not only of interest to producers and consumers but can also have implications for policymaking processes, as the data can help answer fundamental questions, such as "What is the optimum durability of a product?"

For energy-related products regulated under the Ecodesign Directive [13], regulations have been developed, including resource efficiency requirements [36–44]. These regulations are intensively discussed by stakeholders during the policymaking process [45]. The requirements include, among others, durability, availability of spare parts, software updates, and product reparability. The Methodology for the Ecodesign of Energy-related Products (MEErP) considers various economic and ecological aspects [46]. As part of this methodology, life cycle assessments referring to the mass of different materials used to



manufacture an average product (the "base case") are calculated using the EcoReport tool [47]. This tool uses a reference list of conversion factors that must be multiplied by the mass of the materials on the bill of materials (BOM). The results include, among others, $CO_2$ equivalents for each material or material fraction and, as a sum, for the entire product. This reference list is not updated regularly and is currently part of the revision of the entire methodology. In addition, the calculation of the ecological footprint of a product also considers the energy consumption during the use phase. However, in the past, more assumptions were made, and no data collected in situ were used. Instead, theoretical usage profiles and test results under controlled laboratory conditions are often used, which only partially reflect the product's real applications and thus the consumer behavior. Therefore, alternatives or useful additions are required, which the study presented here may lead to. Considering the further shift of the Ecodesign Directive towards the inclusion of resource efficiency as well as the new opportunities presented by IoT, this paper presents a calculation method that allows manufacturers to identify the point of time when it is more efficient (from an economic and ecological perspective) to replace a product rather than to repair it. In the future, it could be applied by policymakers to set up adequate requirements for product durability, availability of spare parts, software and firmware updates for each type of product under one product group and for each type of application. This would allow for more flexible regulation and consider user behavior in a deeper manner.

## 2. Materials and Methods

This section introduces a calculation method for determining the ecologically and economically optimal time to replace a product in use. The application of the calculation method results in a critical durability limit (the durability is expressed as calendar time, number of cycles, etc., related to the product-specific application). If this limit is exceeded, it would be beneficial to replace a product in use with a new product that is likely to be more efficient from an ecological (Section 2.1) or economic (Section 2.2) perspective. Finally, further influences due to aging/wear-out, maintenance, and repair (Section 2.4) are presented.

### 2.1. Environmental Impacts

#### 2.1.1. Input Data

The data required for the application of our calculation method are (i) the bill of materials (BOM) of the related product, (ii) conversion factors (CF) for converting the masses of the materials used into $CO_2$-equivalents (guidance and data are freely or commercially available, e.g., under [46,48,49]) and (iii) the energy consumption of the related product during the use phase. It is necessary to define the units; therefore, either each energy used for production, use, etc., shall be converted into $CO_2$-equivalents, or each $CO_2$-equivalent for each material used shall be converted into an energy unit.

#### 2.1.2. Environmental Impact of a New Product

The environmental impact includes the provision of the materials and energy used to manufacture the product. First, each mass of a specific material $m_{mat,i}$ must be multiplied by the related $CF_{mat,i}$ and the sum calculated. Finally, the energy consumed during the manufacture of the product $E_{process}$ must also be considered. Both the impact of the materials and the production process itself represent the environmental impact of the product $E_{prod}$:

$$E_{prod} = \sum_{i=1}^{n} (m_{mat,i} \cdot CF_{mat,i}) + E_{process} \qquad (1)$$

$n$ = number of different materials of the product.

#### 2.1.3. Environmental Impact of a New Replacing Product

The same procedure as described in the previous subsubsection is applied to an alternative, a new product, which is considered to be a replacement for the product in use. This time, the BOM of the new product, the same reference list of $CF_{mat,i}$ and the

energy used to manufacture the new product $E_{process,repl}$ must be used. In addition, the environmental impact of the product still in use, $E_{prod}$, must be added since the product in use was also manufactured in the past. All two aspects, i.e., the impact of the materials needed for production, the energy consumption of the production process itself, and the impact of the product still in use, represent the environmental impact of the product $E_{prod,repl}$ under consideration for replacement:

$$E_{prod,repl} = \sum_{i=1}^{l}(m_{mat,i} \cdot CF_{mat,i}) + E_{process,repl} + E_{prod} \qquad (2)$$

$l$ = number of different materials of the replacing product.

### 2.1.4. Energy Consumption of a Product in Use

To compare the energy consumption (typically expressed in kWh) of a product during the operation $E_{use}$ with the energy consumption of a new replacing product, two time periods must be included in the calculation: First, the period from the first use of the product $t_0$ to the point of time $t_1$ ($\Delta t_1$), when an alternative product is considered to replace the previously used product. Second, the period from the first time an alternative product is considered to replace the product still in use $t_1$ to a future point of time $t_2$ ($\Delta t_2$). In both periods, the input power $P_{use}$ needs to be considered:

$$E_{use} = \int_{t_0}^{t_1} P_{use}\, dt + \int_{t_1}^{t_2} P_{use}\, dt \qquad (3)$$

### 2.1.5. Energy Consumption of a New Replacing Product

For the energy consumption of the replacing product $E_{repl}$, the input power $P_{repl}$ of this new product shall be considered from the point of time $t_1$ at which the new product is to replace the product still in use, until a future point of time $t_2$. For comparison with the product still in use, the input power of the product still in use $P_{use}$ required over the period from $t_0$ until $t_1$ must be considered since the energy has already been consumed by that point of time $t_1$:

$$E_{use,\,repl} = \int_{t_0}^{t_1} P_{use} + \int_{t_1}^{t_2} P_{repl}\, dt \qquad (4)$$

### 2.2. Economic Impacts

#### 2.2.1. Input Data and Parameters

The input data for the calculation method presented here are the price of the product, the currently valid average electricity price for the consumer, the costs for transport and storage of the product, and, if applicable, for the disposal of the replaced product.

#### 2.2.2. Economic Impact of a Product in Use

The economic impact for the consumer $C_{prod}$ is the price paid at the point of sale. It includes all manufacturers' costs in terms of materials used, salaries, energy consumption in production, transport and storage, marketing, profit margin, sales, etc. Costs do not need to be considered separately, as all costs and profits are reflected in the price of the product.

#### 2.2.3. Economic Impact of a New Product

For a new product to replace a product in use, the same aspects as described in Section 2.2.2 must be considered. In addition to the price paid for the product that is considered a replacement $C_{repl}$, the economic impact of the product still in use $C_{prod}$ must be added since the product in use was also purchased by the same consumer in the past. Therefore, the economic impact of a product intended as a replacement for a product $C_{prod,repl}$ that is in use is the sum of the prices paid by the consumer for both products.

$$C_{prod,repl} = C_{prod} + C_{repl} \qquad (5)$$

### 2.2.4. Running Costs for a Product in Use

The costs for the consumer during the use phase of the product $C_{use}$ must be calculated, as also described in Sections 2.1.4 and 2.1.5, considering two different time spans. First, the time span from the first use of the product still in use $t_0$ to the point in time $t_1$ when an alternative product is considered a replacement for the product still in use. Second, the time span from the first time at which an alternative product is intended to replace the product still in use $t_1$ until a future point of time $t_2$. The result of the application of Equation (3), which describes the energy consumption of a product in use over the previously mentioned time spans, can be directly multiplied by the currently valid average electricity price $c_{pow}$ (consumer price). This results in the costs that the consumer must pay for the operation of the product (additional impacts are discussed in Section 2.4 and are not considered here for simplification):

$$C_{use} = E_{use} \cdot c_{pow} \tag{6}$$

### 2.2.5. Running Costs for a New Product

As described in Section 2.1.5, when calculating the cost for operating a new product as $C_{use,repl}$ a replacement for another product in use, the power consumption at the point of time when the replacement should take place $t_1$ must be taken into account. The results of applying Equation (4) can be directly multiplied by the average energy price $c_{pow}$:

$$C_{use,\,repl} = E_{use,\,repl} \cdot c_{pow} \tag{7}$$

### 2.3. Total Ecological and Economic Impacts

Including all input data from Section 2.1, the total ecological impact of the product still in use can be calculated by applying Equation (8) and that of the product considered a replacement for the product in use can be calculated by applying Equation (9):

$$E_{total} = E_{prod} + E_{use} \tag{8}$$

$$E_{total,repl} = E_{prod,repl} + E_{use,\,repl} \tag{9}$$

Including all input data from Section 2.2, the total economic impact of the product still in use can be calculated by applying Equation (10) and the product that is considered a replacement for the product in use can be calculated by applying Equation (11):

$$C_{total} = C_{prod} + C_{use} \tag{10}$$

$$C_{total,repl} = C_{prod,repl} + C_{use,\,repl} \tag{11}$$

At the point of time $t_{crit,E}$, when $E_{total} = E_{total,repl}$, a product in use and a product considered to replace the product in use, have the same ecological impact. At a point of time later than $t_{crit,E}$, a new product as an alternative to a product in use would lead to a lower ecological impact. Similarly, another point of time $t_{crit,C}$, can be determined. This is shown in Figure 1.

### 2.4. Additional Influences

The performance of the products may change over time, and repair of the products need to be considered. In this subsection, these influences, especially on the parameter $t_{crit}$, are explained.

### 2.4.1. Energy Efficiency as a Function of Time

The required input power of a product is theoretically constant during its lifetime. In this way, the cumulative energy consumed during a certain time increases linearly as a function of time. Any change in energy consumption required to keep the product functional, e.g., due to component wear or loss and aging of lubricants, will gradually have an effect and increase the environmental impact of the product. The environmental

impacts from Section 2.1 do not include wear or fatigue of components, aging of lubricants, etc., to keep the approach presented independently of the type of energy-related product and its technology. However, if such additional influences are considered when applying the presented approach, increasing demand for energy required for the intended function of the product can be considered by a correction factor in Equations (3) and (4). Then the influence of these kinds of effects over time must be determined, described by a function, and this function must be added to the previously mentioned equations. In addition, the point of time of $t_{crit,E}$ depends strongly on the difference between the energy efficiency of both products and the energy used to manufacture the products. The less energy consumed to manufacture the product and the more efficient the product that is considered a replacement for the product in use, the earlier $t_{crit,E}$ occurs.

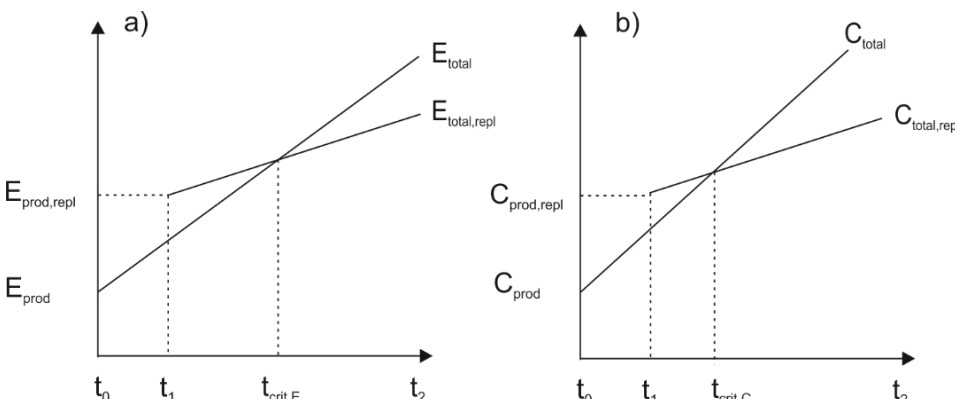

**Figure 1.** Scheme in which the different points of time $t_0$, $t_1$ and $t_2$, the ecological (**a**) and economic (**b**) impacts of the product in use $E_{total}$ and $C_{total}$ and of the product to be considered a replacement for the product in use $E_{total,repl}$, and $C_{total,repl}$ are depicted. $t_{crit,E}$, and $t_{crit,C}$ represent the points of time when the ecological or economic impact of a new product replacing a product in use would be less than the product's impact in use.

### 2.4.2. Repair (and Maintenance)

Both the product in use and the product that is considered a replacement for the product in use must be repaired after certain time spans. For the sake of simplicity, the same number of repair actions, the same relative point of times for repair, and the same type of spare parts are assumed.

The environmental impact of the spare parts, based on the sum of the materials used for production multiplied by the corresponding CF (as described in the first two subsections of this section) and the energy used for the manufacture, storage, and transport $E_{rep}$ of the spare parts must be considered. The contribution of the spare parts to the environmental impact of the product still in use $E_{rep}$ and the product that is to replace the product still in use $E_{rep,repl}$ is equal. However, the spare parts are used at different points of time for the two products, so the contribution to the total environmental impact must be taken into account separately:

$$E_{rep} = \sum_{i=1}^{n}(m_{mat,i} \cdot CF_{mat,i}) + \sum_{k=1}^{m}E_{rep,\,k} + \sum_{i=n+1}^{r}(m_{mat,i} \cdot CF_{mat,i}) + \sum_{k=m+1}^{l}E_{rep,\,k} \qquad (12)$$

$$E_{rep,repl} = 2 \cdot \sum_{i=1}^{n}(m_{mat,i} \cdot CF_{mat,i}) + 2 \cdot \sum_{k=1}^{m}E_{rep,\,k} + \sum_{i=n+1}^{r}(m_{mat,i} \cdot CF_{mat,i}) + \sum_{k=m+1}^{l}E_{rep,\,k} \qquad (13)$$

$n$ = number of different materials used for the original product before $t_1$, $r$ = total number of different materials used, $m$ = number of spare parts of the original product before $t_1$, $l$ = total number of spare parts.

The impact of repair also needs to be reflected in the calculation of the ecological impact, and the Formulas (8) and (9) are extended accordingly:

$$E_{total,\,r} = E_{prod} + E_{use} + E_{rep} \qquad (14)$$

$$E_{total,\,repl,r} = E_{prod,repl} + E_{use,repl} + E_{rep,repl} \tag{15}$$

Analogously, the economic impact on the total economic impact can be calculated:

$$C_{env,use} = C_{prod,total} + C_{use} + C_{rep,use} \tag{16}$$

$$C_{env,new} = C_{prod,new} + C_{new} + C_{rep,new} \tag{17}$$

The points of time at which maintenance and repair are required are often different. Maintenance is carried out at fixed time intervals and is repeated as long as the product is in use (e.g., a filter must be replaced after a certain number of operating hours). This can be done preventively or after the failure of a component that requires frequent maintenance (ink cartridge needs to be refilled in a printer). Therefore, the contribution of product maintenance to the environmental impact and running costs of the entire product is linear on average and is neglected here. In contrast, repair actions are performed on-demand at discrete points of time and are likely to be required more often the longer the product is in use. Consequently, the contribution of the repair to the total impact of the entire product increases as a function of time. If repairs are less necessary the longer the product is in use (rather unlikely), the contribution of the repair to the total impact would decrease over time. Only if the repair actions are carried out at equal time intervals and always the same component must be replaced by the same spare part, the contribution would be linear. In the latter case, the manufacturer would possibly define the process as maintenance rather than repair. If the intervals between the repair actions decrease during the product's lifetime, $t_{crit}$ occurs at an earlier point of time during the lifetime of a product. Accordingly, replacing a product in use with a new product would lead to a lower overall impact of the product at an earlier stage in the product's lifetime. The contributions of the different time intervals of repair are summarized in Figure 2.

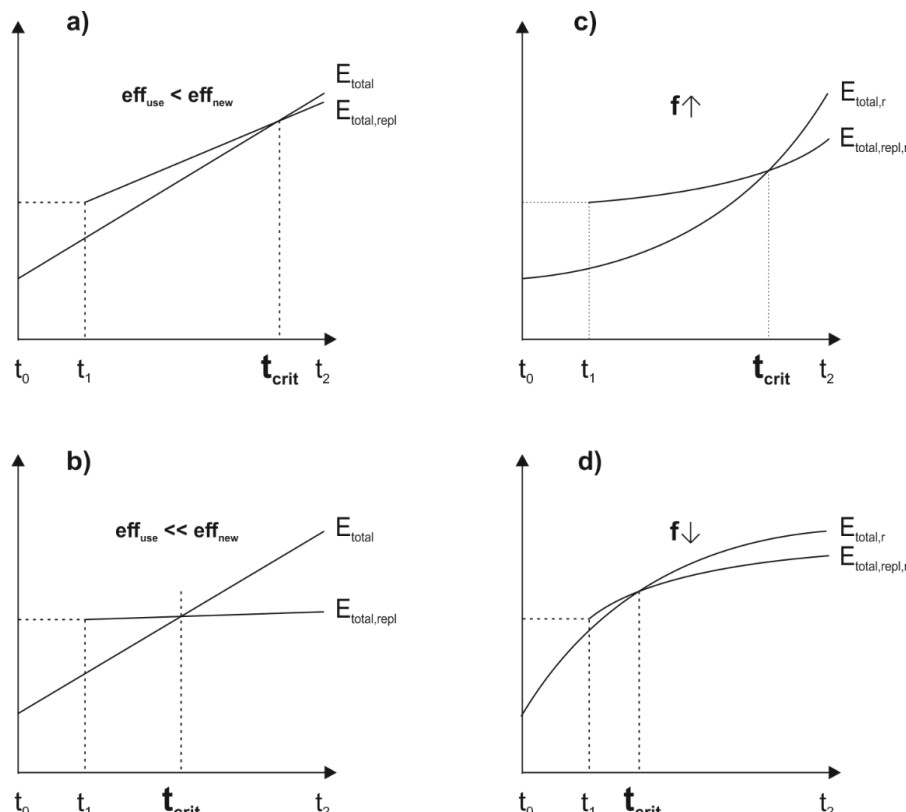

**Figure 2.** Influence of a relatively small (**a**) and large (**b**) difference between the energy efficiencies of both products and influence of increasing (**c**) and decreasing (**d**) frequency of repair actions on $t_{crit}$. To enhance clarity, the curves have been smoothed.

### 3. Results

In the introduction, the potential of IoT and big data was presented from a business perspective, but also for consumers and policymakers. However, the whole concept presented in this study strongly relies on the availability of data on the performance and usage of the devices that need to be retrieved, processed, and analyzed. To take advantage of the opportunities offered by the new paradigm and the calculation method presented here, several challenges [19,50] must be overcome, relating to technology, users, and legal issues regarding security, privacy, and accountability [51].

#### 3.1. Technical Challenges

So far, the smart home market is quite fragmented. This situation makes the real integration and interoperability of heterogeneous systems and technologies [52] difficult. Standards are needed to enable their seamless integration to truly take value from the potential offered by IoT in terms of exploiting the generated and tracked data [19,53]. In addition, the current internet architecture and network capabilities are still too limited in terms of scalability or manageability [52]. There are also challenges in dealing with big data: Adequate data management and mining for the vast amounts of complex, multi-source, heterogeneous, unstructured data are just as necessary, as well as methods and capabilities for analysis to derive value from the opportunities [17,19,54]. Due to the fragmentation of technology and the issues of technology and system integration, we are still far from a truly connected smart home ecosystem, where many users have only a single smart appliance that limits a full value proposition.

#### 3.2. Challenges Related to Consumers/Users

Despite growing familiarity with the technology, setting up and using smart home devices is still not easy—or desirable—for everyone, and studies show that there is still a discrepancy between the number of smart devices sold and the number of those actually used with their smart features. Since many of the appliances, such as dishwashers or washing machines, work well without a connection, users refrain from connecting them to the Internet [14]. While some might simply not appreciate the potential added value, others are reluctant for privacy reasons [16], as they have concerns about data security and integrity. As a result, such issues with vulnerabilities of IoT devices, cybercrime, and attacks need to be solved to support wider adoption [17]. Building trust in smart home appliances and trust in the privacy, safety, and security of user data is a technical, psychological as well as legal challenge facing industry and policymakers.

#### 3.3. Legal Challenges

The calculation of the total impact of a product shall include consideration of the energy consumption during the use phase of the product. Since the products are connected to the Internet (typically wireless) and the data are constantly monitored by third parties, the privacy of consumer data must be protected. In addition, consumers' trust must be enhanced, and consumers must be subsequently motivated to give their consent to share their data. Studies on consumer perception show that consumers are becoming increasingly aware that data is collected and commercialized and that they differentiate their personal data according to who uses the data and what the personal benefit is [55]. The systematic collection and processing of data are acceptable to the consumer at some point, e.g., in smart meters, when consumers can benefit from cost-saving potentials [56]. The General Data Protection (GDPR)/Regulation (EU) 2016/6791 entered into force in the European Union in May 2018. Accordingly, all companies that handle the personal data of EU residents are subject to the GDPR. The GDPR defines the legal basis for the processing of data, which (among other options, Article 6) requires that either the owner of the data has given his consent or that it is the data necessary for the performance of a task in the public interest or in the exercise of official authority. In our case, either the consumer must give his consent (in order to be persuaded), or the consumer must be forced to include the

right of the authorities to collect personal data in an appropriate directive or regulation. A voluntary provision of data would be beneficial, firstly because of the questionable legitimacy of collecting and processing personal data in the interest of resource and energy efficiency and, with this, secondly, to increase the general acceptance of the approach presented in this paper. Furthermore, regarding the question of ownership of personal data in IoT, it needs to be clarified whether the data of connected goods fall under the category of personal or non-personal data [57] since the GDPR defines personal data in Article 4 as "information relating to an identified or identifiable person ("data subject") [ . . . ]. The example of the use of Smart City data [58] raises the question of whether the same logic can also be applied to, e.g., washing machines. Connected appliances are also vulnerable to potential attacks by independent third parties that pose a constant security risk. Among the most popular threat tools are malware or botnets [59] that lead to threats such as denial of service or unauthorized access [2]. Companies would have to ensure that they provide sufficient software updates to ensure the safety and security of their products once they are released on the market [60–63]. So far, however, manufacturers have apparently and repeatedly failed to produce secure products [64], which, according to Moore [65], can be explained by misaligned incentives, information asymmetries, and externalities related to the economics of cybersecurity. To overcome the barriers faced by manufacturers, possibilities such as ex-ante safety regulation, ex-post liability, or disclosure of information have been proposed [65]. Certification based on standards as underlying requirements has been identified by European policymakers as one instrument to address this issue [66], with the EU Cybersecurity Act (Regulation (EU) 2019/881) "introducing for the first time an EU-wide cybersecurity certification framework for ICT products, services and processes" [67]. In view of the approach presented in this paper, companies offering connected products should, therefore, be encouraged to take appropriate steps towards enhancing cybersecurity. Certification can, therefore, help to build consumer trust based on the signaling theory [68] and encourage consumers to give their consent to sharing their data.

## 4. Discussion

IoT leads to innovative and individual product applications. In this context, the legal requirements affecting these products may also need to be reconsidered. Hence, product design studies of manufacturers have required that the environmental conditions and use profiles of products be assumed when they are in use by consumers. However, this information is used, among other things, as input data for testing and simulating the product's durability, future need for maintenance, repair, and additional services [69]. In parallel, there are requirements for placing a product on the market. These requirements are also developed based on average product models (representing whole product groups or sectors of product groups), standard environmental conditions, and use patterns. These simplifications of the real use conditions lead to relatively nonspecific product designs and legal requirements. With the availability of large amounts of real product data associated with IoT, opportunities arise to improve and optimize both product design and legal requirements. Optimized product design based on data generated by increased connectivity is widely discussed in the literature [7–12]. What is missing, however, are discussions about optimized regulatory requirements taking advantage of the new data available through IoT.

The calculation method presented in this paper allows the product requirements to be defined for each product individually, considering the environmental conditions under which the product performs its function. It represents a fundamental shift from the current approach, where product requirements have fixed values, e.g., for the availability of spare parts or software and firmware updates and durability. Fixed values do not accurately represent all various products, usage profiles, and environmental conditions. Setting requirements for a relative point of time (when a product in use is less efficient than a possible new product) as an alternative for an absolute point of time (as practiced in policymaking today) would be much more effective. In addition, the efficiency of an entire

population of a product can be determined online in the case that a representative number of consumers are willing to share their data.

$t_{crit}$ is the factor that could be used as a reference point for the policymaking process and is, therefore, the scope of the paper presented here—the following requirements could be improved or set, respectively:

1. By using the method presented in this paper, the manufacturer would be able to calculate that replacing a product in use with a new product sold on the market at a comparable price would lead to ecological and economic advantages. In a connected world, the manufacturer may be in a legal position to no longer support the product, e.g., no longer supply software updates or spare parts—which would lead to a phase-out of the products from the market. The conditions under which the availability of spare parts, software updates, etc., is mandatory can be set by policymakers, e.g., defining that a product must be supported by the manufacturer until at least XY% of the products in use are less efficient compared to new products of the same brand and price category. This would make the regulation of reparability of energy-related products more flexible and efficient. It is certainly up to the user, as the owner of the energy-related product, to decide whether to continue using the device after this point has been reached;

2. The calculation is described above is only possible by assuming that the new product would last until the $t_{crit}$ is reached. This means that the durability of a new product is higher than the interval between $t_1$ and $t_{crit}$. In case a manufacturer declares that a product replacement would be more ecological and/or economical and no longer needs to be supported, the manufacturer must state the expected durability of the new product (at least in the declaration of conformity). The policymaker may not regulate the durability itself but may set an obligation to repair the product(s) during the stated durability. This would also make the regulation of the durability of energy-related products more flexible.

In summary, legal requirements can be set that must be fulfilled until after a defined number of products have reached a certain state—instead of regulating fixed time spans. This would lead to a very flexible product regulation that addresses any application and even environmental condition.

## 5. Conclusions

From a political point of view, a "planned obsolescence" as a requirement of the Ecodesign Directive could be an effective, flexible and dynamic approach to set requirements for the availability of product information, software updates, spare parts, etc., as well as for setting minimum durability requirements. This would effectively improve the circularity of energy-related products, as mentioned and supported, e.g., in the European Commission's circular economy action plan [35]. In addition, the data from smart products will enable an easier prediction of spare parts needs and software and firmware updates. Stocks of spare parts can be adapted to the actual demand, ensuring availability according to legal requirements. Not only technical feasibility in terms of connectivity and data gathering is essential for the functioning of the calculation method presented in this paper. It also requires the consumer's acceptance and consent so that this product can be digitally monitored and information on his or her behavior (concerning the use of the products) can be transmitted to the manufacturer. Data privacy and security issues must be taken seriously. Time will tell whether the issues discussed in this paper are the future perspective of policymaking in a smart world where the product works. In any case, smart products operating within the scope of the IoT should also be regulated smartly.

**Author Contributions:** Conceptualization, M.-C.S.; Methodology, M.-C.S. and A.H.; Writing—original draft, M.-C.S., C.K. and M.M.; Writing—review & editing, M.-C.S., C.K., M.M. and A.H. All authors have read and agreed to the published version of the manuscript.

**Funding:** This research received no external funding.

**Institutional Review Board Statement:** Not applicable.

**Informed Consent Statement:** Not applicable.

**Data Availability Statement:** Not applicable.

**Conflicts of Interest:** The authors declare no conflict of interest.

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
