# Peer review of "Smart Products Enable Smart Regulations—Optimal Durability Requirements Facilitated by the IoT"

_sustainability, doi:10.3390/su13084395_

Round 1

Reviewer 1 Report

This manuscript has well research framework and design. This manuscript presents a method for calculating the economically and ecologically optimal durability of a product. The research model determined the point in time when a product should be replaced by combining consumer data with product design data. However, the following comments are listed below:

  1. The references format could be revised to fulfill journal template.
  2. There is no title in the Y axis in the figure 2 (a)(b)(c)(d). Please add the title name in the figure 2.
  3. All of the codes should be explained clearly in all equations and formula.

Author Response

Dear reviewer,

Thank you for your comments and suggestions for improvement. Below you will find our responses to your comments:

1.The references format could be revised to fulfill journal template.

Improved

2. There is no title in the Y axis in the figure 2 (a)(b)(c)(d). Please add the title name in the figure 2.

The title of the y-axis is different for each curve. For better comparability, the titles are depicted on the curves directly.

3. All of the codes should be explained clearly in all equations and formula.

Explanations of the variables are given in the related text above the formulas including the kind of data, units and, where appropriate, within the caption of the formula.

Kind regards,

Moritz-C. Schlegel

Reviewer 2 Report

The aim of the paper is very interested but some reviews are needed to improve the overall quality:

  1. I suggest to review the title, it is not so very well aligned with the content of the paper. 
  2. IoT seems to be a relevant and fundamental point of the study. Literature about previous studies on economical and/or ecological impacts calculation in this field is missed. It is important to reinforce the analysis of studies in a similar field in order to clarify the innovativeness of your contribution.
  3. Seection 1.1 is not so concentrated on product lifecycle management. "Big data and product lifecycle management" is a huge topic under discussion by several scholars. I suggest to review the title and aim of this sub-section or to improve the contents.
  4. You have used different sections for defining input data but you should improve these sections adding further details. For example you could introduce all the variables and related initials for providing a unique point to check all the variables used in the remaining part of the paper.
  5. For some companies can be difficult to collect the data for all the specified variables, such as for E. Problems for data collection could be discussed in the paper.
  6. The contents of the result section are more appropriate for the discussion one.
  7. I suggest to review the style of your sections' titles. For example you use italic for section and sub-section.
  8. In the paper you usually use the word Chapter. The use of chapter for different parts of a scientific paper is not so common. It is usually preferred section and sub-section.

Author Response

Dear reviewer,

Thank you for your comments and suggestions for improvement. Below you will find our responses:

1. I suggest to review the title, it is not so very well aligned with the content of the paper.

Improved

2. IoT seems to be a relevant and fundamental point of the study. Literature about previous studies on economical and/or ecological impacts calculation in this field is missed. It is important to reinforce the analysis of studies in a similar field in order to clarify the innovativeness of your contribution.

Improved

3. Section 1.1 is not so concentrated on product lifecycle management. "Big data and product lifecycle management" is a huge topic under discussion by several scholars. I suggest to review the title and aim of this sub-section or to improve the contents.

Improved

4. You have used different sections for defining input data but you should improve these sections adding further details. For example you could introduce all the variables and related initials for providing a unique point to check all the variables used in the remaining part of the paper.

Explanations of variables are provided in the associated text, including the type of data, the units required, and the relationship between variables. Some variables are explained in the formula label, if appropriate. 

5. For some companies can be difficult to collect the data for all the specified variables, such as for E. Problems for data collection could be discussed in the paper.

The bill of materials is prepared by the manufacturers. Sources for conversion factors are given.

6. The contents of the result section are more appropriate for the discussion one.

Agreed, it would also fit the discussion. However, since a methodology is presented, we preferred to present here the challenges of application as a result.

7. I suggest to review the style of your sections' titles. For example you use italic for section and sub-section.

Improved

8. In the paper you usually use the word Chapter. The use of chapter for different parts of a scientific paper is not so common. It is usually preferred section and sub-section.

Improved

Kind regards,

Moritz-C. Schlegel

Round 2

Reviewer 2 Report

The paper has been improved, I suggest its publication.